# Association between the Phase Angle and the Severity of Horizontal Gaze Disorder in Patients with Idiopathic Dropped Head Syndrome: A Cross-Sectional Study

**DOI:** 10.3390/medicina59030526

**Published:** 2023-03-08

**Authors:** Ryunosuke Urata, Tatsuya Igawa, Shomaru Ito, Akifumi Suzuki, Norihiro Isogai, Yutaka Sasao, Haruki Funao, Ken Ishii

**Affiliations:** 1Department of Orthopaedic Surgery, School of Medicine, International University of Health and Welfare, Chiba 286-8520, Japan; 2Department of Rehabilitation, International University of Health and Welfare Mita Hospital, Tokyo 108-8329, Japan; 3Department of Physical Therapy, School of Health Science, International University of Health and Welfare, Tochigi 324-8501, Japan; 4Department of Rehabilitation, International University of Health and Welfare Narita Hospital, Chiba 286-8520, Japan; 5Department of Orthopaedic Surgery, Keio University School of Medicine, Tokyo 160-8582, Japan; 6Society for Minimally Invasive Spinal Treatment (MIST), Tokyo 101-0063, Japan

**Keywords:** dropped head syndrome, horizontal gaze disorder, phase angle, body composition

## Abstract

*Background and Objectives*: The phase angle, calculated by bioelectrical impedance analysis, can help elucidate the pathology of patients with idiopathic dropped head syndrome (IDHS) and explain the motor dysfunction associated with the horizontal gaze disorder. The aim of this study was to clarify the characteristics of phase angle in IDHS patients and the relationship between the phase angle and the severity of horizontal gaze disorder. *Materials and Methods*: This cross-sectional study included 43 female patients with IDHS and 69 healthy female volunteers. A multi-frequency segmental body composition analyzer was used to calculate body composition parameters, including whole-body and lower extremity phase angles. Propensity score (PS) matching analysis was performed to compare the body composition parameters between the IDHS and healthy groups. Variables that determine the PS were identified by correlation analysis, using the whole-body phase angle as the dependent variable. In addition, correlation analysis was performed between the severity of horizontal gaze disorder as assessed by McGregor’s slope (McGS), phase angle, and other body composition parameters. *Results*: Unadjusted group comparisons showed no significant difference in whole-body and lower extremity phase angles between the IDHS and healthy groups. PS matching created a total of 38 matched pairs for age, height, and fat-free mass index. Although the comparison between groups of matched samples showed no significant difference in the whole-body phase angle, the lower extremity phase angle in the IDHS group was significantly lower than that in the healthy group (*p* = 0.033). Correlation analysis showed significant negative correlations only between McGS and whole-body (*r* = −0.31, *p* = 0.043) and lower extremity phase angle (*r* = −0.39, *p* = 0.009) in the IDHS group. *Conclusions*: Abnormal body composition of the lower extremities were observed in IDHS patients. Furthermore, it was suggested that horizontal gaze disorder in IDHS patients is associated with whole-body and lower extremity phase angles.

## 1. Introduction

Dropped head syndrome is more common in older women [1], and the ability to maintain horizontal gaze is reduced due to neck deformity. Regarding reports on incidence rate, a former systematic review encompassing 74 studies documented the existence of 129 cases [1], and a multicenter, retrospective, cohort study conducted across three Japanese centers disclosed 67 cases within the timeframe of 2014 to 2018 [2]. Albeit the worldwide incidence rate, the disease remains indeterminate, and it is plausible that it may surge in the forthcoming years, given the aging demographic. Etiology of dropped head syndrome is diverse and some cases are idiopathic, without a primary disease such as Parkinson’s disease and myasthenia gravis [1]. Although the cause of idiopathic dropped head syndrome (IDHS) is unclear, patients are known to present with motor deficits extending to the lower extremities. According to previous reports, IDHS patients show abnormal posture during standing and walking [3,4], have a slower gait than healthy subjects [3], and the severity of horizontal gaze disorder is associated with the patient’s gait speed [5]. Furthermore, while there have been reports on surgical treatment [6] and rehabilitation [7] for the treatment of IDHS, no effective treatment modality has been established. However, some case series and interventional studies reported that IDHS symptoms improved due to programs that include lower extremity or whole-body exercises [7,8,9]. This evidence suggests that musculoskeletal attributes of patients might be related to the etiology and symptoms of IDHS.

The use of bioelectrical impedance analysis (BIA) can easily and safely estimate the body composition of the whole body and each body part [10], and is useful for understanding the pathology of patients. Previous studies that evaluated the body composition of patients with IDHS using BIA have described the soft tissue mass of the patients. According to Eguchi et al. [11], the lean mass and skeletal muscle mass of the extremity were lower in patients than in healthy controls, whereas Igawa et al. [12] reported conflicting results. The body composition in patients with IDHS is not yet well understood, as implied by these inconsistent results.

In recent years, the phase angle calculated by BIA has attracted attention as a marker associated with disease prognosis and physical status of elderly patients [13]. A higher phase angle value indicates healthier cellularity [13]. Previous studies reported that the phase angle is associated with skeletal muscle mass and muscle quality in healthy adult and elderly subjects [14,15]. Yamada et al. [16] reported that the lower extremity phase angle was associated with aerobic capacity, complex gait ability, and total fitness age score in the elderly, better than that of the whole body and arms. Furthermore, walking speed in the elderly is associated with muscle mass and phase angle, in addition to muscle quality of the lower extremities [17]. Both the quantity and quality of the muscle should be evaluated for impairment of physical function in the elderly. However, to the best of our knowledge, there are no reports that include data regarding muscle quality in body composition analysis of patients with IDHS. Although studies using phase angle have the potential to forward current knowledge of the pathology of IDHS, there are no studies that have investigated this.

We hypothesized that whole-body and/or lower extremity phase angles in IDHS patients are lower than that in healthy elderly subjects, and that these phase angles would be associated with the severity of horizontal gaze disorder. This is because IDHS patients have motor impairments extending to the lower extremities, and the reduced motor function of patients is associated with horizontal gaze disorder [4,5,7]. An analysis of phase angles in patients may clarify the pathology of IDHS and will provide useful information for planning the development of therapeutic algorithms and treatment programs. The purposes of this study were (1) to compare whole-body and lower extremity phase angles in IDHS patients with that of healthy subjects, and (2) to clarify the relationship between the whole-body and lower extremity phase angles, and horizontal gaze disorder in IDHS patients.

## 2. Materials and Methods

### 2.1. Participants

This cross-sectional study consecutively recruited female patients with IDHS, admitted to a university hospital in the Kanto region of Japan between June 2019 and April 2022. Those with metal implanted in the body, a history of lower extremity or spine surgery, autoimmune disease, inflammatory disease, chronic kidney disease, malignant tumor, difficulty standing, and those who refused evaluation were excluded. The IDHS diagnosis criteria were defined with reference to previous studies [18]. The criteria were as follows: (1) failure to maintain horizontal gaze while standing and walking; (2) cervical deformity is passively correctable; and (3) no obvious comorbidities associated with the dropped head syndrome. Spine surgeons examined the patients using their physical findings and radiographs, and patients received a differential diagnosis from neurologists as needed.

Sixty-nine healthy female volunteers living in the Kanto region of Japan between June 2019 and February 2020 were recruited for the control group. Female volunteers were recruited by the municipal public relations department.

### 2.2. Ethical Considerations

The study was conducted according to the guidelines of the Declaration of Helsinki and approved by the Institutional Review Board of the International University of Health and Welfare (IRB#5-19-20 and #18-Io-158-2, Institutional date, June 2019).

### 2.3. Data Collection

#### 2.3.1. Body Composition

Body composition data were calculated by BIA using a multi-frequency segmental body composition analyzer (MC-780A; Tanita Co., Ltd., Tokyo, Japan). All participants had their body composition measured before exercise or at least 3 hours after eating. Values for muscle mass, fat body mass, and lean body mass were respectively calculated for the whole body, appendage, trunk, and lower extremities. Body mass index (BMI) and fat-free mass index (FFMI) were calculated by dividing the respective weight and lean body mass values by the square of the participant’s height. The phase angle was calculated using the values of resistance ® and reactance (Xc) when a current of 50 kHz was applied. Phase angle was calculated using the following formula [13]:phase angle (degree) = [−arctangent (Xc/R) × 180°/π]

#### 2.3.2. Severity of Horizontal Gaze Disorder

The severity of horizontal gaze disorder was measured by McGregor’s slope (McGS), using standing lateral radiographs. McGS was defined as the angle of the line from the posterior aspect of the hard palate to the opisthion, in relation to the horizontal line [19]. Previous studies have suggested high validity and reliability for the assessment of gaze using the McGS [19,20].

### 2.4. Statistical Analysis

Results were presented as the means (standard deviation (SD)). SPSS version 25.0 (IBM, Armonk, NY, USA) was used for all statistical analyses. The Shapiro–Wilk test was used to assess the normality of each dataset. Propensity score (PS) matching was used for adjusting covariates in both groups [21]. Previous studies have reported covariate relationships between phase angle and age, height, and FFMI [22,23]. However, some researchers have stated that BMI is a determinant of phase angle [13,24]. In our study, the covariate was determined after evaluating the respective correlations between phase angle and FFMI and BMI in both groups. A logistic regression model was used to estimate the PS. One-to-one sampling without replacement was created using the nearest neighbor method. The caliper distance was set to 0.20 times the SD of PS logit. Considering that no previous studies have verified the phase angle of IDHS patients, in our study, the necessary sample size was verified by using post hoc power analysis. The power analysis used G*Power 3.1 (Heinrich Heine University, Düsseldorf, Germany). The Student’s *t*-test and Mann–Whitney U test were used for comparison of the groups. Pearson’s product-rate correlation coefficient and Spearman’s rank correlation coefficient were used for evaluating the correlations between whole-body phase angle and BMI and FFMI in both groups, and between McGS and the attribute and body composition variables in the IDHS group. A *p*-value < 0.05 was considered statistically significant.

## 3. Results

A flowchart of patient selection in each group is shown in Figure 1. From a total of 78 female patients with IDHS, we evaluated 73 for eligibility after excepting those who refused to participate. Excluding those with metal implanted in the body (*n* = 6), a history of lower extremity and spine surgery (*n* = 4), autoimmune disease (*n* = 4), inflammatory disease (*n* = 3), chronic kidney disease (*n* = 4), malignant tumor (*n* = 6), and difficulty standing (*n* = 3), a total of 43 patients were included in this study. No missing data were included in both the IDHS and healthy groups.

A comparison of demographic and body composition data between both groups is shown in Table 1. Height in the IDHS group was significantly higher than that in the healthy group (*p* < 0.01), and body weight and BMI in the IDHS group were significantly lower than that in the healthy group (*p* < 0.05, *p* < 0.001). The McGS value was 14.0 ± 20.3 degrees. Muscle mass and lean body mass in each body part, other than the trunk, were significantly greater in the IDHS group than that in the healthy group, and fat body mass in all body parts in the IDHS group was significantly less than that in the healthy group (Table 1). There were no significant differences between groups in terms of whole-body and lower extremity phase angles (*p* = 0.65, *p* = 0.50).

Correlation analysis showed a significant positive correlation between whole-body phase angle and FFMI in both groups (IDHS group, *r* = 0.43, *p* = 0.004; healthy group, *r* = 0.65, *p* < 0.001) (Figure 2). Although the correlation between whole-body phase angle and BMI was statistically significant in the healthy group, it was not significant in the IDHS group (IDHS group, *r* = 0.11, *p* = 0.49; healthy group, *r* = 0.41, *p* < 0.001) (Figure 3). Therefore, age, height, and FFMI were included in the logistic regression model for estimating PS.

Using the PS matching method to compare each variable adjusted by age, height, and FFMI value, 38 participants in both groups were selected (Table 2). There was no significant difference between the two groups adjusted for age, height, and FFMI (*p* = 0.73, *p* = 0.64, *p* = 0.82). BMI in the IDHS group was significantly lower than that in the healthy group, and fat body mass in all body parts in the IDHS group was significantly lower than that in the healthy group (Table 2). Although there was no significant difference between groups in terms of whole-body phase angle (*p* = 0.72), the lower extremity phase angle in the IDHS group was significantly lower than that in the healthy group (*p* = 0.033).

From the relationship of body composition data in the statistical correlation analysis, McGS was revealed to be significantly correlated with the whole-body phase angle (*r* = −0.31, *p* = 0.043) (Table 3, Figure 4A) and lower extremity phase angle (*r* = −0.39, *p* = 0.009) (Table 3, Figure 4B). There was no significant correlation detected between McGS and the other variables. As a result of post hoc power analysis (two tales, effect size ρ = 0.39, type I error probability = 0.05, sample size = 43), the power was 77% (i.e., type II error probability = 0.23), and sufficient detection power was obtained for the correlation between McGS and lower extremity phase angle in the IDHS group.

## 4. Discussion

The purpose of this study was to compare whole-body and lower extremity phase angles of patients with IDHS with those of healthy controls, and to clarify the association between whole-body and lower extremity phase angles and horizontal gaze disorder in patients with IDHS. To the best of our knowledge, there are no reports on the analysis of phase angles in patients with IDHS. A comparison of whole-body and lower extremity phase angles of IDHS patients with healthy controls showed that their lower extremity phase angle values were lower than those of the controls. In addition, correlation analysis showed significant negative correlations between the whole-body and lower extremity phase angles and the severity of horizontal gaze disorder assessed by McGS in IDHS patients.

The phase angle is based on the value of two actions (reactance (Xc) and resistance (R)) that resist the alternating current applied to the organism [13]. Theoretically, reactance in the human body has reflected the function of a capacitor by the cell membrane, and water in soft tissues has been recognized as a resistor [25]. Therefore, the phase angle represents the distribution of water between the intracellular and extracellular spaces, and a high-phase angle depends on a low extracellular water/intracellular water ratio [26]. The lower extremity phase angle of patients with IDHS was significantly lower than those of healthy controls in this study. This result showed partial support for our hypothesis. One reason for the difference in lower extremity phase angle between both groups may be related to the fat body mass. A previous study reported that phase angle shows a tendency to increase with BMI up to a BMI of 35 kg/m^2^ in a population older than 70 years of age [27]. This can be thought of as reflecting an increase in muscle and fat cell number and/or size. Since the muscle mass in the two groups was comparable in this study, an increase in fat cell number and/or cell size in the lower extremities may be associated with an increase in phase angle. However, although the fat body mass of all body parts in IDHS patients was lower than that of controls, the difference in phase angle values was detected only in the lower extremities. A possible explanation for this is that there is a difference in fluid distribution between patients and controls. A previous study in heart failure patients showed that the presence of peripheral edema was associated with the variability of phase angle [28]. In addition, a study in hemodialysis patients reported that increased edema decreased the phase angle, regardless of fat or muscle mass [29]. In this study, chronic kidney disease and malignancy that can cause peripheral edema have been excluded. However, varicose veins frequently occur in older women [30]. A previous study reported that female patients with spinal deformities have a higher prevalence of varicose veins [31]. Therefore, it seems necessary for the analysis to focus on fluid distribution in the lower extremities, such as venous hemodynamics, to explain the difference in phase angle between the IDHS and the controls.

The phase angle is associated with the physical status of the elderly. As we hypothesized, there was a significant negative correlation between the whole-body and lower extremity phase angles, and the McGS in IDHS patients. This shows that as the gaze drops further in patients with dropped gazes, the phase angle becomes lower. A previous cross-sectional study reported an association between the severity of the McGS and decreased walking speed and cervical type deformity in IDHS patients [5]. In addition to these variables, this study suggested that the whole-body and lower extremity phase angles were related to McGS. A previous study reported that the phase angle is associated with physical function battery and muscle quality in the elderly [14,15,16,32]. The results of this study suggest that IDHS symptoms are closely associated with the patient’s whole-body and lower extremity muscle function. Clinicians should, thus, include whole-body and lower extremity phase angles and motor function assessments in the patient’s clinical findings, and closely observe the medical condition. Furthermore, according to a report by Igawa et al. [3] that compared the dynamic alignment during walking of IDHS patients with healthy subjects, the ankle joint dorsiflexion angle during gait was higher in IDHS patients than in healthy subjects, and the plantar–flexion movement of the ankle joint was lower in IDHS patients than in healthy subjects. As a cause of the inefficient activity of the ankle plantar–flexor muscles, dysfunction of the calf muscle may be involved. Dysfunction of the calf muscle is frequently observed in venous disease patients [33,34,35], and the ankle plantar–flexor strength and gait speed are lower in chronic venous disease patients than in healthy controls [35]. Considering the possibility that the phase angle is associated with patient hemodynamics and fluid distribution, decreased lower extremity phase angle values in IDHS patients may be associated with calf muscle dysfunction. A more detailed study is warranted to clarify the relationship between muscle function and phase angle in IDHS patients.

Our study suggested an association between the onset of IDHS and the phase angle. The body composition of IDHS patients differed from that of controls, and the lower extremity phase angle was lower than that of controls. Clinicians and researchers should therefore include the lower extremity phase angle as a variable in the analysis of IDHS pathology. In addition, this discussion may be further advanced by including a section on fluid distribution of the patient’s lower extremities. As another result of our study, it was revealed that the reduction of whole-body and lower extremity phase angles was associated with the severity of horizontal gaze disorder. Body composition measurement, using BIA, is useful in that it is easily and noninvasively performed [10]. Therefore, both clinicians and patients can safely obtain the data. To provide useful data for decision making, researchers may be able to analyze whether the phase angle can predict treatment outcomes or prognosis.

Our study has some limitations. First, because this was a cross-sectional study, we cannot infer the causal relationship between IDHS and phase angle, and between phase angle and horizontal gaze disorder in IDHS patients. Second, the IDHS patients included in our study were recruited from a single institution and were females only. Gender is one of the determinants of the phase angle [13,36]; therefore, the results of our study cannot be extrapolated to all IDHS patients. A multi-institutional study, including both males and females, should be planned in the future. Third, because our study included hospitalized patients, the results may have been biased toward a patient population with higher severity of horizontal gaze disorder. Finally, our study has been unable to clearly explain the differences in lower extremity phase angle between patients and controls. Bioelectrical impedance vector analysis (BIVA) is a recently developed technique that can obtain information about the patient’s hydration status, body cell mass, and cell integrity [13]. It is possible that differences in body composition between patients and controls could be analyzed in more detail by using BIVA.

## 5. Conclusions

Our analysis of phase angles in IDHS patients revealed that the lower extremity phase angle in IDHS patients was lower than that in healthy elderly. Furthermore, the phase angle of the whole-body and lower extremities of IDHS patients demonstrated a significant correlation with the severity of horizontal gaze disorder. Our findings suggested that deciphering anomalous fluid distribution in the lower extremities may furnish a more intricate comprehension of the patient’s pathology. Moreover, it has also been proposed that phase angle may be leveraged for treatment and prognosis predictors. To investigate whether phase angle influences treatment decision-making, results of longitudinal studies are needed in the future.

## Figures and Tables

**Figure 1 medicina-59-00526-f001:**
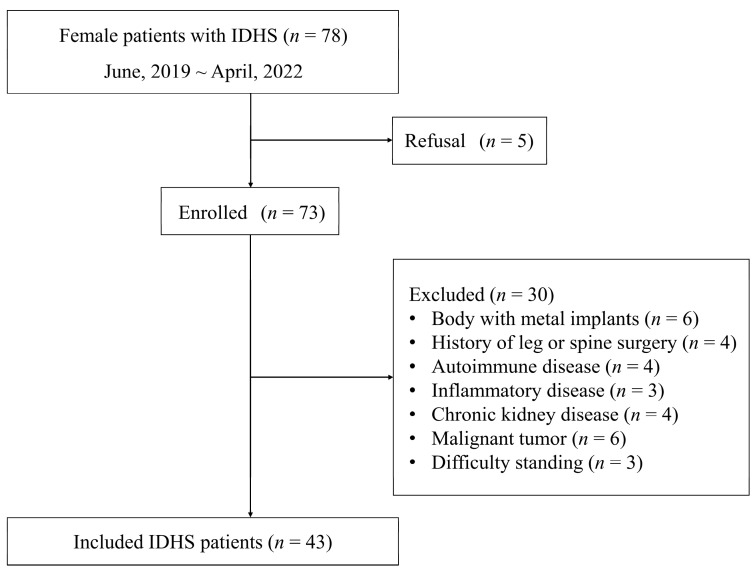
Flowchart of patient selection.

**Figure 2 medicina-59-00526-f002:**
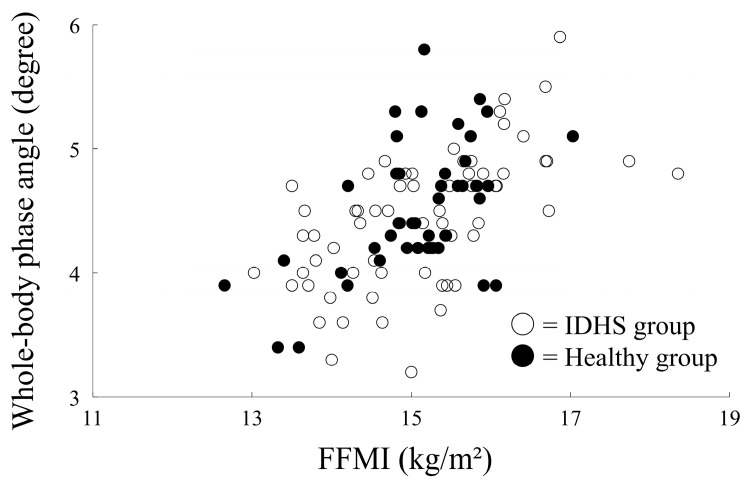
Correlation between FFMI and phase angle in both groups.

**Figure 3 medicina-59-00526-f003:**
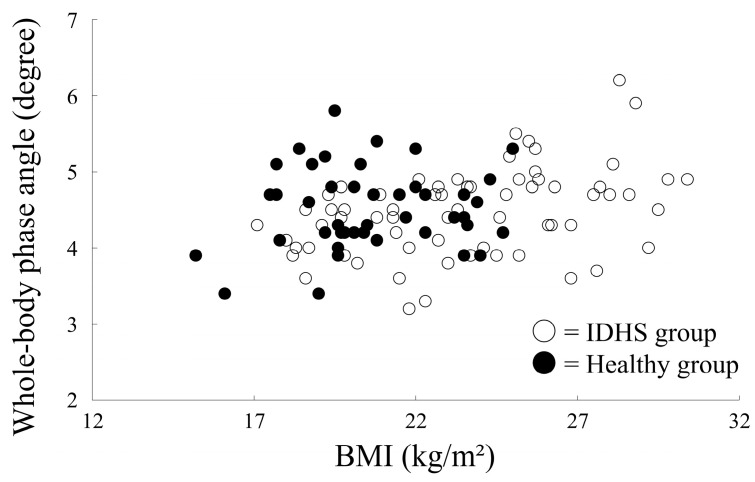
Correlation between BMI and phase angle in both groups.

**Figure 4 medicina-59-00526-f004:**
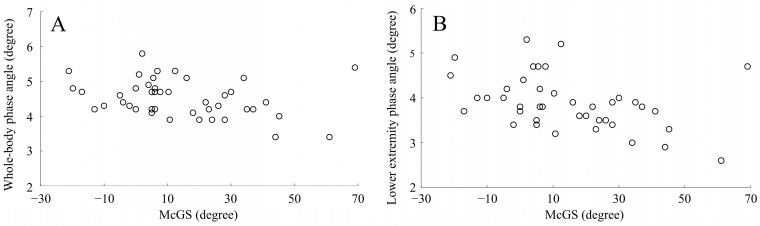
Correlation between McGS and phase angles in the IDHS group. (**A**) Correlation between McGS and whole-body phase angle; (**B**) correlation between McGS and lower extremity phase angle. Abbreviations: McGS, McGregor’s slope.

**Table 1 medicina-59-00526-t001:** Demographic data and results of comparison between IDHS and healthy groups.

Variables	IDHS (n = 43)	Healthy (n = 69)	*p*-Value
Age, years	74.3 (6.4)	78.6 (6.9)	0.082
Height, m	1.51 (0.06)	1.48 (0.07)	0.011
Mass, kg	47.5 (7.2)	51.6 (9.1)	0.031
BMI, kg/m^2^	20.6 (2.4)	23.5 (3.4)	<0.001
FFMI, kg/m^2^	15.1 (0.8)	15.2 (1.2)	0.71
McGS, degree	14.0 (20.3)	-	-
Duration of symptoms, month	25.1 (27.4)	-	-
Body composition			
Muscle mass, kg	32.8 (3.3)	31.8 (4.4)	0.024
Fat body mass, kg	12.8 (5.1)	18.1 (5.1)	<0.001
Lean body mass, kg	34.7 (3.5)	33.5 (4.7)	0.026
Appendicular muscle mass, kg	14.6 (2.2)	13.7 (2.7)	0.010
Appendicular fat mass, kg	5.7 (2.1)	8.1 (2.5)	<0.001
Appendicular lean mass, kg	15.5 (2.3)	14.5 (2.9)	0.044
Trunk muscle mass, kg	18.2 (1.2)	18.1 (1.9)	0.14
Trunk fat mass, kg	7.1 (3.3)	10.0 (3.5)	<0.001
Trunk lean mass, kg	19.2 (1.3)	19.0 (2.1)	0.12
Lower extremity muscle mass, kg	11.8 (1.8)	10.6 (2.2)	0.004
Lower extremity fat mass, kg	4.7 (1.8)	6.6 (2.0)	<0.001
Lower extremity lean mass, kg	12.5 (2.0)	11.2 (2.4)	0.005
Total body water, kg	25.3 (2.9)	25.1 (3.6)	0.76
Whole-body phase angle, degree	4.5 (0.5)	4.5 (0.5)	0.65
Lower extremity phase angle, degree	3.9 (0.6)	4.0 (0.7)	0.50

All results are reported as the mean (standard deviation). Abbreviations: IDHS, idiopathic dropped head syndrome; BMI, body mass index; FFMI, fat-free mass index; McGS, McGregor’s slope.

**Table 2 medicina-59-00526-t002:** Group comparisons using data for PS matching.

Variables	IDHS (n = 38)	Healthy (n = 38)	*p*-Value
Age, years	77.2 (6.3)	76.7 (6.5)	0.73
Height, m	1.51 (0.06)	1.50 (0.06)	0.64
BMI, kg/m^2^	20.5 (2.4)	23.0 (3.8)	0.008
FFMI, kg/m^2^	15.1 (0.9)	15.0 (1.1)	0.82
Muscle mass, kg	32.8 (3.1)	32.1 (3.7)	0.39
Fat body mass, kg	12.4 (5.1)	18.0 (6.5)	<0.001
Lean body mass, kg	34.4 (3.6)	33.9 (4.0)	0.25
Appendicular muscle mass, kg	14.4 (2.2)	14.0 (2.1)	0.44
Appendicular fat mass, kg	5.6 (2.0)	8.2 (2.7)	<0.001
Appendicular lean mass, kg	15.4 (2.2)	14.9 (2.3)	0.27
Trunk muscle mass, kg	18.1 (1.2)	18.1 (1.8)	0.44
Trunk fat mass, kg	7.1 (3.3)	9.8 (3.9)	<0.001
Trunk lean mass, kg	19.2 (1.2)	19.0 (1.9)	0.27
Lower extremity muscle mass, kg	11.6 (1.9)	11.0 (1.7)	0.13
Lower extremity fat mass, kg	4.6 (1.7)	6.7 (2.1)	<0.001
Lower extremity lean mass, kg	12.4 (1.9)	11.6 (1.8)	0.08
Total body water, kg	25.3 (2.6)	25.1 (3.5)	0.86
Whole-body phase angle, degree	4.5 (0.5)	4.5 (0.5)	0.72
Lower extremity phase angle, degree	3.8 (0.6)	4.1 (0.6)	0.033

All results are reported as the mean (standard deviation). Abbreviations: IDHS, idiopathic dropped head syndrome; BMI, body mass index; FFMI, fat-free mass index.

**Table 3 medicina-59-00526-t003:** Correlation analysis between McGS and the attribute, and body composition variables in the IDHS group.

Variables (n = 43)	*r*	*p*-Value
Age, years	0.06	0.68
Height	0.18	0.25
Mass	−0.05	0.76
BMI	−0.20	0.19
FFMI	−0.10	0.54
Muscle mass	0.16	0.31
Fat body mass	−0.12	0.45
Lean body mass	0.15	0.34
Appendicular muscle mass	0.16	0.32
Appendicular fat mass	−0.18	0.26
Appendicular lean mass	0.10	0.54
Trunk muscle mass	0.12	0.46
Trunk fat mass	−0.07	0.67
Trunk lean mass	0.12	0.44
Lower extremity muscle mass	0.12	0.43
Lower extremity fat mass	−0.17	0.28
Lower extremity lean mass	0.12	0.43
Total body water	0.02	0.92
Whole-body phase angle	−0.31	0.043
Lower extremity phase angle	−0.39	0.009

Abbreviations: BMI, body mass index; FFMI, fat-free mass index.

## Data Availability

The data presented in this study are available upon reasonable request from the corresponding authors. The data are not publicly available due to privacy restrictions.

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
