# Peer review of "Association between the Phase Angle and the Severity of Horizontal Gaze Disorder in Patients with Idiopathic Dropped Head Syndrome: A Cross-Sectional Study"

_medicina, 2023, doi:10.3390/medicina59030526_

Round 1

Reviewer 1 Report

Introduction: Kindly provide effected number of population either in US or worldwide. 

What are some of the current techniques used to treat people with the mentioned condition? 

What is the novelty of this study? 

How is the second part of the aim addressed in the results and conclusion section? Kindly elaborate more. 

Methods: How was the sample size calculated?

Author Response

We would like to thank the Reviewer 1 for your comments. Please see the attachment.

Reviewer 2 Report

I would like to request Urata et al. to work on following points while resubmitting this manuscript to the journal. 

1) Kindly improve all 3 tables and all 4 figures.  The figures and tables should be bright and clear.

Author Response

We would like to thank the Reviewer 2 for your comment. Please see the attachment.
